

# The interplay between movement, morphology and dispersal in *Tetrahymena* ciliates

Frank Pennekamp[1,2], Jean Clobert[3] and Nicolas Schtickzelle[1]

[1] Earth and Life Institute & Biodiversity Research Centre, Université Catholique de Louvain, Louvain-la-Neuve, Belgium
[2] Department of Evolutionary Biology and Environmental Studies, University of Zurich, Zurich, Switzerland
[3] Station d'Ecologie Théorique et Expérimentale, CNRS, Moulis, France

## ABSTRACT

Understanding how and why individual movement translates into dispersal between populations is a long-term goal in ecology. Movement is broadly defined as 'any change in the spatial location of an individual', whereas dispersal is more narrowly defined as a movement that may lead to gene flow. Because the former may create the condition for the latter, behavioural decisions that lead to dispersal may be detectable in underlying movement behaviour. In addition, dispersing individuals also have specific sets of morphological and behavioural traits that help them coping with the costs of movement and dispersal, and traits that mitigate costs should be under selection and evolve if they have a genetic basis. Here, we experimentally study the relationships between movement behaviour, morphology and dispersal across 44 genotypes of the actively dispersing unicellular, aquatic model organism *Tetrahymena thermophila*. We used two-patch populations to quantify individual movement trajectories, as well as activity, morphology and dispersal rate. First, we studied variation in movement behaviour among and within genotypes (i.e. between dispersers and residents) and tested whether this variation can be explained by morphology. Then, we addressed how much the dispersal rate is driven by differences in the underlying movement behaviour. Genotypes revealed clear differences in terms of movement speed and linearity. We also detected marked movement differences between resident and dispersing individuals, mediated by the genotype. Movement variation was partly explained by morphological properties such as cell size and shape, with larger cells consistently showing higher movement speed and higher linearity. Genetic differences in activity and movement were positively related to the observed dispersal and jointly explained 47% of the variation in dispersal rate. Our study shows that a detailed understanding of the interplay between morphology, movement and dispersal may have potential to improve dispersal predictions over broader spatio-temporal scales.

Corresponding author
Frank Pennekamp,
frank.pennekamp@ieu.uzh.ch

## INTRODUCTION

Individual movement is a universal feature of life with broad implications for the ecology and evolution of species (*Turchin, 1998*). As most environments are spatially structured, understanding how individuals move across increasingly fragmented landscapes is of crucial importance (*Baguette & Van Dyck, 2007*). Individual movement can be defined as 'any change in the spatial location of an individual in time' (*Nathan et al., 2008*). Dispersal movements are more specifically defined as the result of a specific movement type, that is movement that can potentially (but does not necessarily) lead to gene flow (*Baguette, Stevens & Clobert, 2014*) and are vital for the persistence of spatially-structured populations. Although dispersal implies a change in spatial position, it goes beyond mere movement: it is a central life history trait (*Bonte & Dahirel, 2017*), which can be conceptualised as a three stage process where decisions are taken during emigration, transition and immigration (*Clobert et al., 2009*). Movement patterns may hence vary according to the costs of dispersal (*Bonte et al., 2012*), for instance due to the type of habitat that is encountered (*Schtickzelle et al., 2007*). Few studies try to integrate drivers of small-scale individual movements with dispersal, although previous work has shown the potential of movement to predict large scale spatial dynamics from short spatio-temporal scales, if variation in movement is properly accounted for (*Morales & Ellner, 2002*). This is important because dispersal has wide implications for population dynamics and the spatial distribution of genetic diversity (*Bowler & Benton, 2005*; *Ronce, 2007*; *Clobert et al., 2012*; *Jacob et al., 2015a*).

Variation in movement and dispersal, and covariation with traits such as morphology and behaviour, is the raw material for selection in spatially structured environments and can lead to dispersal syndromes, that is, consistent co-variation among traits (*Ronce & Clobert, 2012*; *Stevens et al., 2012*). Variation in both movement and dispersal has been reported within and among many different organisms (*Austin, Bowen & McMillan, 2004*; *Mancinelli, 2010*; *Chapperon & Seuront, 2011*; *Ducatez et al., 2012*; *Debeffe et al., 2014*; *Dahirel et al., 2015*). Some of this variation can be due to environmental causes (e.g. different resource availability, *Fronhofer et al., 2018*), but there is also evidence for genetic effects (*Haag et al., 2005*; *Edelsparre et al., 2014*). As only the latter can lead to the evolution of dispersal and movement strategies, it is important to understand when dispersal and movement variation is genetically or environmentally based.

The development of new technology has recently given us a better grasp on how individual variation in movement is related to dispersal. Individual tracking of roe deer showed that exploratory movements were mainly performed by individuals that would later disperse (*Debeffe et al., 2013*, *2014*), and butterflies show links between movement ability and dispersal (*Stevens, Turlure & Baguette, 2010*). Currently, effects of proxies like body condition are very species and context-specific. However, movement traits have potential to more generally predict which individuals are most likely to disperse.

Besides movement, differences in morphology, physiology and behaviour have been found when comparing dispersers and residents (*Niitepõld et al., 2009*; *Edelsparre et al., 2014*). For instance, body condition and morphology have been found to influence

individual dispersal decisions in mole rats, ciliates, lizards and butterflies and many other organisms (*O'Riain, Jarvis & Faulkes, 1996*; *Fjerdingstad et al., 2007*; *Clobert et al., 2009*; *Stevens et al., 2012*; *Turlure et al., 2016*). Body size is another important predictor of movement, and has been shown to directly influence the speed with which animals can move (*Hirt et al., 2017a*, *2017b*). In general, larger animals can move faster, however, the relationship is non-linear with an optimum, suggesting that the largest species are not necessarily the fastest.

Linking individual movement to dispersal requires us to characterise and understand the underlying sources of variation in both, which has so far mostly be done on insects (*Niitepõld et al., 2009*; *Edelsparre et al., 2014*). Assessing dispersal and movement simultaneously is difficult because dispersal events (especially long-distance) are difficult to track in the field, and recording movement behaviour with adequate resolution and sample size is technically challenging, leading to the use of indirect methods (*Flaherty, Ben-David & Smith, 2010*). Alternatively, relationships between dispersal and movement ability have been studied across taxonomic groups in a comparative fashion (*Dahirel et al., 2015*). One noteworthy exception using a direct approach is a study that investigated and supported links between phenotypic and genotypic differences in larval food foraging and dispersal as adults in *Drosophila melanogaster* (*Edelsparre et al., 2014*). 'Rover' larvae tend to move longer distances and may leave food patches when foraging, whereas 'sitters' tend to move less and rest within their food patch (*Osborne et al., 1997*). In dispersal assays the 'rover' genotype also moved greater distances as adult flies, highlighting genetic links between larval mobility and adult dispersal (*Edelsparre et al., 2014*). Experiments with microscopic organisms are ideal to study the connections between dispersal and movement experimentally, because they allow tight control of the genetic and environmental context and hence allow these to be disentangled.

Experimental approaches with microscopic organisms are a convenient way to measure movement and dispersal simultaneously and hence allow us to study pattern and process at a relevant spatial scale (*Menden-Deuer, 2010*; *Kuefler, Avgar & Fryxell, 2012*). Moreover, controlled experiments can partition how much variation in movement is due to genetic and non-genetic sources and therefore advance our understanding of the mechanistic underpinnings of movement strategies and their evolution. In this study, we used the microbial *Tetrahymena thermophila* experimental system.

There is compelling evidence that dispersal in this organism is not solely a diffusive process, but depends on individual decisions triggered by environmental cues. Previous work has revealed that cells modify their dispersal decisions according to cooperative strategies (*Chaine et al., 2010*; *Jacob et al., 2016*), conspecific density and density proxies (*Pennekamp et al., 2014*; *Fronhofer, Kropf & Altermatt, 2015*), social information from conspecifics (*Jacob et al., 2015b*) as well as competition (*Fronhofer et al., 2015*), and perform adaptive habitat choice according to thermal preferences (*Jacob et al., 2017*, *2018*). Extensive variation in dispersal has previously been observed among genotypes of this actively moving ciliate, however, the underlying movement processes have remained elusive.

Previous work has revealed extensive variation in life history traits among genotypes, including trade-offs in general growth performance (including high dispersal ability) and formation of specialised dispersal morphs (*Fjerdingstad et al., 2007*). Later work also revealed dispersal plasticity regarding conspecific density, which could be partly explained by morphological differences (body size and shape) among genotypes (*Pennekamp et al., 2014*).

In this study, we investigate the relationships between small-scale individual movement (i.e. cell trajectories), dispersal (i.e. emigration rate) and morphological features (i.e. body size and shape) across 44 genotypes of *T. thermophila*. We characterised the movement behaviour in terms of activity (number of actively moving cells) and quantitative movement behaviour (speed and the characteristic scale of velocity autocorrelation) via video-based cell tracking (*Pennekamp, Schtickzelle & Petchey, 2015*). In addition, we measured morphological properties of each genotype, as well as its dispersal rate across the two-patch system. With this data, we addressed the following questions:

1. Is there variation in movement behaviour within genotypes (between dispersers and residents) and among genotypes?
2. Can this movement variation be explained by morphology (cell size and shape)?
3. How much is the dispersal rate driven by differences in the underlying movement behaviour (activity and movement differences among genotypes)?

## MATERIALS AND METHODS

### Model organism

*Tetrahymena thermophila* is a 30–50 µm long unicellular, ciliated protozoan inhabiting freshwater ponds and streams in the eastern part of North America, where it naturally feeds on patches of bacteria and dissolved nutrients (*Doerder & Brunk, 2012*). We used a set of 44 genetically distinct genotypes (clonally reproducing as isolated lines) differing in several life history traits (*Fjerdingstad et al., 2007*; *Schtickzelle et al., 2009*; *Chaine et al., 2010*; *Pennekamp et al., 2014*). All genotypes are stored in suspended animation (frozen in liquid nitrogen) and can be ordered from the Tetrahymena Stock Center (https://tetrahymena.vet.cornell.edu/). Genotypes were kept as isolated monocultures in 'common garden' conditions over a large number of generations (>100) after defrosting, under axenic conditions in Proteose peptone medium enriched with yeast extract, at constant 27 °C in a light controlled incubator with a 14:10 h light/dark cycle both prior and during the experiment. Refer to Supplemental Information 1 for additional information on these genotypes and details of culture conditions.

### Experimental quantification of dispersal and movement parameters

We quantified dispersal rate and movement parameters of *T. thermophila* cells using a fully factorial experimental design implying two factors of interest: the genotype (44 genotypes) and the dispersal status (dispersers vs. residents). We used the same standardised two-patch system (subsequently referred to as dispersal system) developed in previous

work (*Fjerdingstad et al., 2007*; *Schtickzelle et al., 2009*; *Chaine et al., 2010*; *Pennekamp et al., 2014*), consisting of two 1.5 mL microtubes connected by a silicon pipe (internal diameter four mm, tube length 17 mm), filled with medium (see Fig. S1). To start the experiment, cells of a single genotype were pipetted into the 'start' tube to obtain a density of 300,000 cells/mL, an intermediate cell density commonly observed under our culturing conditions. After mixing the medium to distribute cells evenly in the start tube and 30 min of acclimation, the connecting pipe was opened, and cells could freely disperse. At the end of the experiment after 6 h, the pipe was closed by a clamp and five independent samples were taken from both the start and the target tubes of each dispersal system. Cells found in the 'start' or 'target' are subsequently referred to 'residents' or 'dispersers,' respectively, the two modalities possible for the dispersal status variable. Five dark field images (one for each chamber; resolution: 5,616 × 3,744 pixels) and one 40 s long video (of a randomly chosen chamber; HD resolution: 1,920 × 1,080 pixels; 25 frames per second) were then taken using a Canon EOS 5D Mark II mounted on a Nikon Eclipse 50i microscope with a 4× lens; the real size of the imaged area is about 6.3 × 4.5 mm and was not bounded by external borders, hence cells could swim in and out the viewing field. Supplemental Information 1 gives additional information about the experimental protocol and material used.

Images were treated using an objective and automated image analysis workflow to count individual cells and record morphology descriptors (*cell size* and *cell shape*); this workflow is based on ImageJ (*Schneider, Rasband & Eliceiri, 2012*) and was carefully validated and extensively optimised to produce accurate and repeatable results (*Pennekamp & Schtickzelle, 2013*). *Dispersal rate* of a genotype was estimated as the ratio of density in the target tube to the overall density (start + target), that is the proportion of cells in the target.

Individual cell trajectories were obtained from the digital videos in a standardised and automated fashion with a workflow that was later transformed into the R package BEMOVI (*Pennekamp, Schtickzelle & Petchey, 2015*) and was successfully used in previous studies extracting movement characteristics from video sequences (*Banerji et al., 2015*; *Fronhofer, Kropf & Altermatt, 2015*; *Griffiths et al., 2018*). The position of each cell was followed over all the 1,000 frames (40 s long video with 25 frames per second; Fig. S2). First, the *activity* level of cells was computed from videos as the ratio of cells that moved (trajectory duration >1 s and minimum displacement >50 μm, i.e. one body length) divided by the total number of trajectories (moving and non-moving).

Then, trajectories were analysed with continuous time movement models (*Fleming et al., 2014*; *Gurarie et al., 2017*) to compute movement speed and linearity. Continuous time movement models are a natural choice for high-frequency sampling of video microscopy because they can deal with autocorrelation in the movement speed and positions. We used the smoove package in R (*Gurarie et al., 2017*) to fit a hierarchical family of correlated velocity models, basically continuous-time equivalents of the widely applied correlated random walk, with biologically intuitive parameters such as movement speed and the velocity autocorrelation timescale (a measure of the decay in directional persistence). For each genotype, we randomly subsampled 23 trajectories per replicate and tube resulting in a total of 6,072 trajectories. The subsampling was necessary because

analysis with continuous time movement models is computationally demanding due to the model selection procedure involved. Subsampling also ensured the same number of data points per genotype. For each trajectory, we fitted four models: an unbiased correlated velocity model (UCVM), an advective correlated velocity model (ACVM), a rotational correlated velocity model (RCVM) or a rotational advective correlated velocity model (RACVM). The best fitting model for a given trajectory was selected via a model selection procedure based on the Akaike information criterion (AIC), and parameters of the model estimated. For each trajectory, we extracted two parameters for further analysis: the *movement speed* (in root mean square) and the velocity autocorrelation timescale (parameter tau), essentially a measure of *movement linearity*. When tau tends towards zero, the movement approaches random Brownian motion, while tau tending towards infinity indicates perfect linear motion (*Gurarie et al., 2017*). We used the velocity likelihood fitting method rather than the exact fitting procedure implemented in smoove, because smoove currently supports the exact fitting approach for the UCVM model only. To check the robustness of the approximate fitting, we performed a check that indicated a negligible bias towards lower movement speed when using the approximate fitting (Fig. S3). We therefore proceeded with the approximate fitting approach. Before further analysis, we performed an outlier exclusion based on the Median Absolute Deviation with a threshold of three (*Leys et al., 2013*) for the two parameters estimated. The Supplemental Information 1 gives additional details concerning trajectory reconstruction from video, cleaning and estimation of movement metrics.

In summary, each dispersal system produced measures for six response variables: two morphology descriptors (cell *size* and *shape*, extracted from images), three movement descriptors (*activity*, *speed*, and *linearity* extracted from videos), and *dispersal rate* (computed from cell densities extracted from images). For all statistical analyses, these response variables were aggregated to produce two values per dispersal system, one for the start tube (residents) and another for the target tube (dispersers); indeed, the true level of replication in this experiment was the dispersal system (genotype × dispersal status combination) and not the individual trajectory. With three dispersal systems (replicates) per genotype, sample size was 264 (44 genotypes × 3 replicates × 2 dispersal status); note that one dispersal system (genotype 32_I) was discarded due to a technical failure of the dispersal system, meaning $n = 262$. Cell size and shape were averaged over all cells found on the five images recorded per tube; activity was directly measured at the video level (one measure per tube) and hence already 'pre-aggregated' at the correct level; speed and linearity were averaged over the 23 trajectories analysed by continuous time movement models on each video; and dispersal rate was computed from densities averaged over the five images recorded per tube.

## Statistical analyses

To address our first question, activity and movement metrics (speed and linearity) were compared among genotypes and among dispersal status (disperser vs. resident cells) using a three-way ANOVA, with genotype and dispersal status as crossed and fixed effects, and replicate as random effect nested in genotype but crossed with dispersal status.

Genotype was considered as a fixed effect, despite its common consideration as a random effect (*Crawley, 2007*). This is because the set of genotypes cannot be considered as a random sample of the genetic variation exhibited by the species in the wild (some genotypes were selected due to previous results or based on their phenotypic characteristics, some others were created by inbreeding in the laboratory). Dispersal status was crossed with replicate because the data for the two statuses (disperser and resident, i.e. target and start tubes respectively) were paired for each dispersal two-patch system. Speed and linearity (tau) were ln-transformed to improve normality of residuals.

All cells belonging to the same genotype should have the same genetic make-up; however, environmental differences encountered during the cell life cycle may lead to different morphologies and cell states. Therefore, to answer our second question, we tested whether differences in movement behaviour between residents and dispersers may be explained by morphological differences such as cell size and shape. To see if there were differences between residents and dispersers, we built ANCOVA models that related movement speed and linearity to morphology properties (size and shape) across genotypes, accounting for differences due to dispersal status. As some of the observed variation may be due to variation across replicates, we investigated how within replicate differences in morphology affect differences in movement. We used the AIC to determine the most parsimonious model, that is the simplest model (in terms of number of parameters) within 2 units (deltaAIC < 2) of the best model (i.e. with the lowest AIC).

To address our third question about the power of movement behaviour to predict dispersal rate, we assessed how much variation in dispersal rate was explained by genotype-specific activity, movement speed, movement linearity and all predictors together. We used the $R^2$ of a multiple regression and compared the three models with the AIC to find the best fitting model. For this analysis, movement metrics (activity, movement speed and linearity) were averaged at the genotype level, that is, over dispersers and residents.

## RESULTS

### Q1: Variation in movement behaviour within and among genotypes

Model selection across the four types of correlated velocity models revealed that the ACVM was the most common across genotypes, indicating the genotypes show directed movement. The dispersal status did not change the overall pattern, but genotypes showed variation in the relative frequencies of movement models (Fig. 1). Genotypes differed in activity (minimum 39% to maximum 70% of total cell population moving) and movement parameters extracted from the correlated velocity models: movement speed (minimum 75 to maximum 289 µm/s) and linearity (tau: minimum 0.039 to maximum 0.13). Additionally, a highly significant difference was shown between dispersal status: compared to residents, dispersers were characterised by a higher activity (0.62 ± 0.05 vs. 0.57 ± 0.08) and faster and more linear movements (speed ± SD: 171 ± 52.5 µm/s vs. 139 ± 52.0; tau: 0.0804 ± 0.0271 vs. 0.0602 ± 0.0244). For the majority of genotypes the dispersers moved faster and more linear, while for some genotypes the opposite was

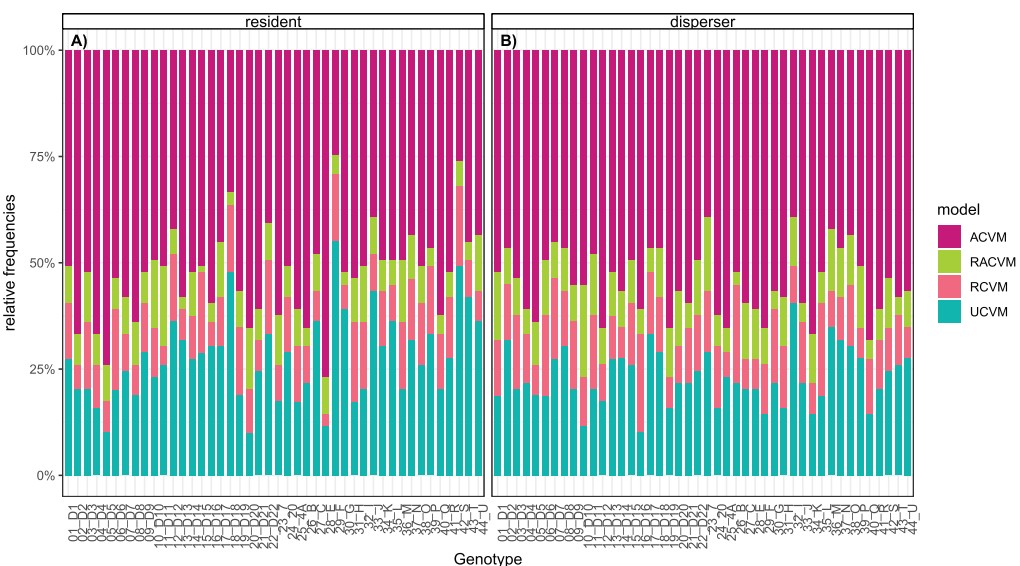

**Figure 1 Model selection for the four types of continuous time movement models fitted to 23 randomly selected trajectories per genotype.** Relative frequencies of the most parsimonious model shown for (A) resident trajectories across genotypes (B) disperser trajectories across genotypes. The ACVM model is the most represented, followed by the UCVM. Some trajectories are best represented by rotational variants (RACVM and RCVM).

**Table 1 Three-way ANOVA to assess the effect of genotype and the dispersal status (i.e. difference between dispersers and residents) on three movement metrics: activity (proportion of moving cells), movement speed and linearity.** Genotype and dispersal status were considered as crossed and fixed effects, and replicate as random effect nested in genotype but crossed with dispersal status because data from the two status were paired per replicate (i.e. the start and target tubes of one dispersal system). The column 'denominator for *F*-test' indicates the error term used to test for each effect, according to the ANOVA model; '–' denote the factors that cannot be tested because the error has no degrees of freedom in this model.

| Response variable | | | Activity | | | | Speed: ln (speed) | | | | Linearity: ln (tau) | | | |
|---|---|---|---|---|---|---|---|---|---|---|---|---|---|---|
| Factor | Denominator for *F*-test | DF | SS | MS | *F* value | *p* | SS | MS | *F* value | *p* | SS | MS | *F* value | *p* |
| Genotype | Replicate (genotype) | 43 | 0.872 | 0.020 | 2.88 | <0.0001 | 24.927 | 0.580 | 12.40 | <0.0001 | 24.666 | 0.574 | 7.50 | <0.0001 |
| Dispersal status (disperser vs. resident) | Replicate × dispersal status (genotype) | 1 | 0.186 | 0.186 | 42.88 | <0.0001 | 3.193 | 3.193 | 149.28 | <0.0001 | 6.718 | 6.718 | 93.19 | <0.0001 |
| Genotype × dispersal status | Replicate × dispersal status (genotype) | 43 | 0.445 | 0.010 | 2.39 | 0.0003 | 3.977 | 0.092 | 4.32 | <0.0001 | 7.036 | 0.164 | 2.27 | 0.0006 |
| Replicate (genotype) | Error | 87 | 0.612 | 0.007 | – | – | 4.067 | 0.047 | – | – | 6.653 | 0.076 | – | – |
| Replicate × dispersal status (genotype) | Error | 87 | 0.377 | 0.004 | – | – | 1.862 | 0.021 | – | – | 6.272 | 0.072 | – | – |
| Error | na | 0 | 0 | | – | | 0 | | – | | 0 | | – | |
| Total | | 261 | 2.490 | | | | 38.020 | | | | 51.317 | | | |

observed (significant genotype × dispersal status interaction for both movement metrics; Table 1; Fig. 2). Across genotypes the speed and linearity strongly positively co-varied ($b = 0.000383$, $t = 10.961$, $p < 0.001$), meaning faster cells also swam straighter. Neither intercept nor slope differed between residents and dispersers (Fig. S4).

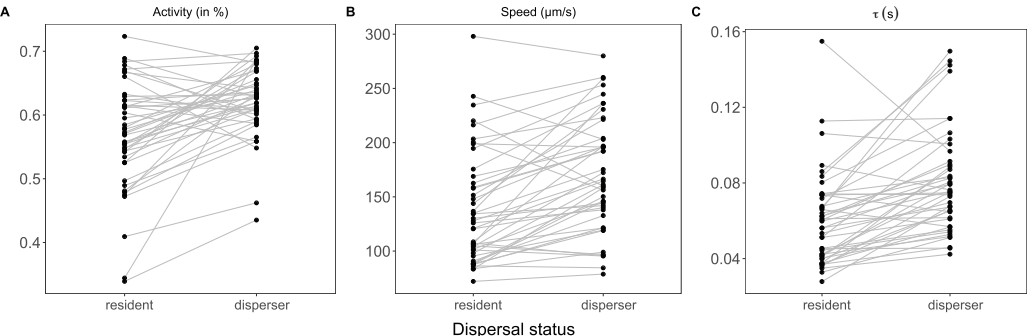

**Figure 2 Overview of among and within genotype variation in (A) activity, (B) speed and (C) tau, that is, linearity.** Each line shows a genotype and its slope indicates differences in movement among status (disperser vs. resident).

## Q2: Link between movement behaviour and morphology

First, the influence of cell morphology on cell movement across genotypes and replicates was analysed (Fig. 3). The most parsimonious model indicated a positive effect of size on movement speed in addition to the higher speed generally found in dispersers (Table S2). Speed was also affected by shape differences: more elongated disperser cells moved faster, whereas the opposite was observed for residents (Table S2). We also found that larger cells moved straighter. The slope of this relationship did not differ among dispersal status, however, dispersers moved straighter on average (Table S3).

The relationship between shape and linearity again was dependent on the dispersal status: whereas higher elongation led to more linear movement for dispersers, residents showed no pattern with higher elongation (Table S3). Within genotypes, larger relative size of dispersers compared to residents led to higher relative movement speed, whereas a larger relative elongation resulted in a decrease in relative speed (Fig. S5; Tables S4 and S5).

## Q3: Predicting dispersal rate based on movement parameters

Consistent with previous experiments, we observed major differences among genotypes in dispersal rate in the two-patch experiment (Fig. 4). The genotypes had significantly different dispersal rates over 6 h (one-way ANOVA: $F_{43, 87} = 9.93$, $p < 0.001$), continuously distributed in the 7–71% range; the majority of genotypes had a dispersal rate lower than 50%. Variation among the 44 genotypes in activity and movement behaviour explained a substantial amount of the variation observed in their dispersal rates. Only considering activity explained 27% of the variation in dispersal rates among genotypes (AIC = −56.21). The genotype-specific movement linearity explained a lower amount of variation (24%, AIC = −54.55) while speed explained a larger percentage of the dispersal variation (37%, AIC = −62.86). Including activity, speed and linearity explained almost 50% of the variation in dispersal (47%, AIC = −66.79). This result indicates that activity and movement features jointly influence the dispersal rate exhibited by a genotype and provide complementary information about dispersal.

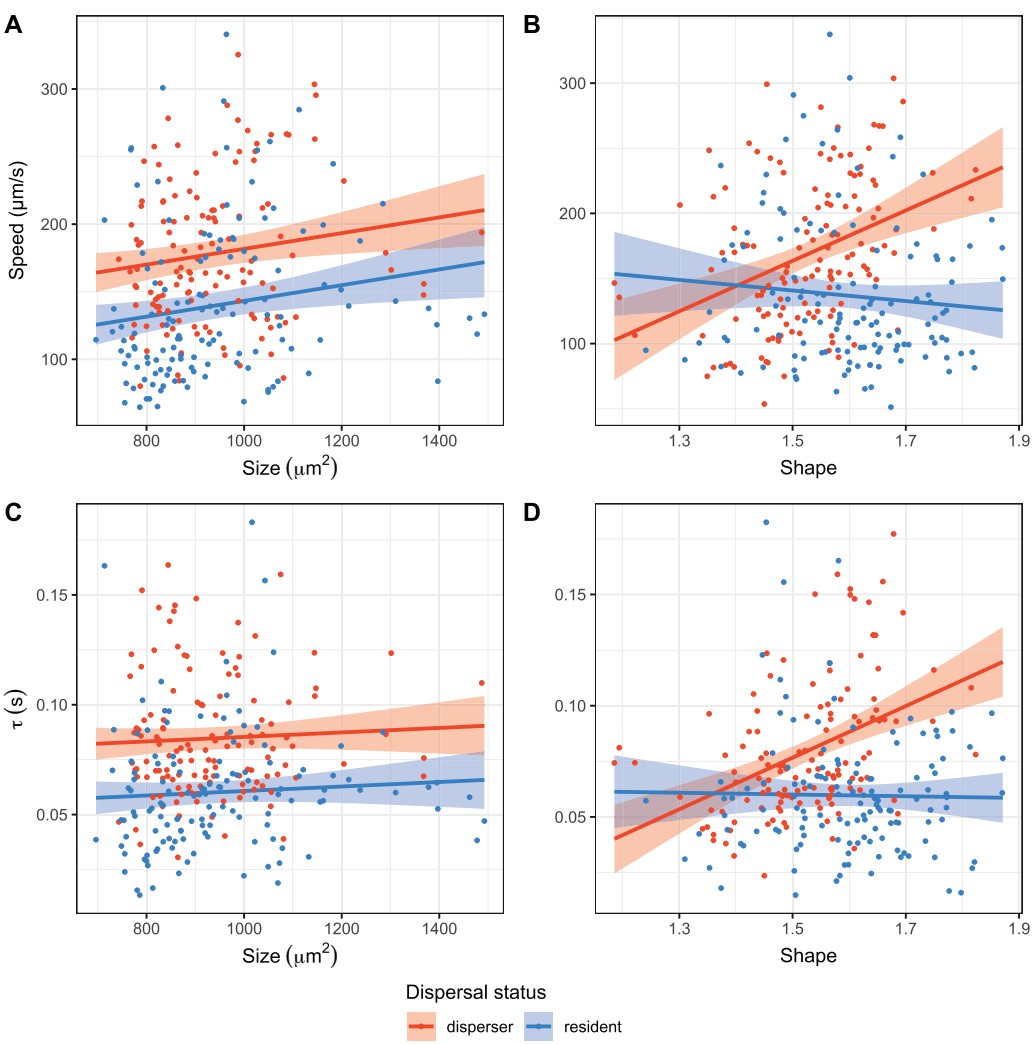

**Figure 3 Relationships between speed (A and B) and tau, that is, linearity (C and D), dispersal status (red and blue) and cell morphology (size and shape).** Lines and confidence intervals show the partial effects of size and shape of the most parsimonious ANCOVA model ($n$ = 262). Larger cells moved faster but not more linear, with an overall higher level in dispersing cells. In contrast, only in dispersing cells elongation resulted in faster and straighter movement, whereas the opposite was observed in resident cells.

## DISCUSSION

We show that 44 genotypes of *T. thermophila* kept in 'common garden' conditions over many generations exhibit continuous variation in movement parameters (activity, movement speed and linearity). Activity, movement speed and linearity were found to be genotype-dependent, and differed with dispersal status. Although cells within the same genotype have the same genetic make-up, environmental differences encountered during the cell life cycle may lead to different movement behaviours. We show that some of the movement variation can indeed be explained by morphological differences among genotypes and this may explain also within genotype variation. Finally, movement

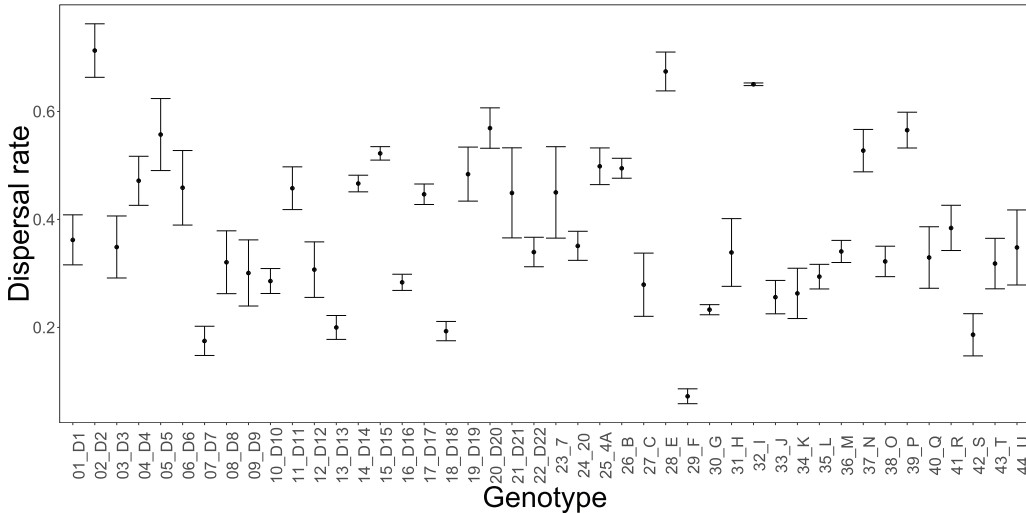

**Figure 4** **The 44 genotypes differed in their dispersal rate in the two-patch experimental system over a period of 6 h.** The point represents the mean dispersal and the error bars the standard error of the mean ($n = 3$ per genotype, except 32_I where $n = 2$).

variation and cell activity was highly predictive of dispersal, explaining 47% of the observed variation.

## Genotype-based movement behaviour differences

So far there are a limited number of model systems where the genetic basis of dispersal has been studied in detail (*Wheat, 2012*). In *Drosophila*, allelic variation in the candidate gene *for* is known to influence the foraging behaviour of larvae; additionally recent research has demonstrated that phenotypic and genotypic variation mainly due to the *for* gene also influences adult dispersal distances (*Edelsparre et al., 2014*). Interestingly, the protein encoded by the *for* gene in *Drosophila*, a cGMP-dependent protein kinase, responsible for the observed behavioural variation in foraging, is also known to influence cilia-mediated chemotaxis in *T. thermophila* (*Leick & Chen, 2004*). Another example is the nematode *Caenorhabditis elegans* where the *npr1* gene is associated with both foraging strategy and dispersal behaviour (*Gloria-Soria & Azevedo, 2008*). Finally, dispersal is heritable in the butterfly *Melitaea cinxia* on the Aland archipelago: young and isolated populations have higher frequencies of dispersive female individuals carrying the *PGI* genotype, a genotype associated with higher flight metabolic rate that increases the probability to reach such habitats (*Haag et al., 2005*). These examples show that genetic links between movement and dispersal exist and are consistent with our results, where movement over short spatio-temporal scales correlates with dispersal over much larger spatio-temporal scales. *T. thermophila* may be a good model species for studying these questions using experimental evolution approaches. Promising directions for future research would be to understand how different selection pressures for movement (within patches) and dispersal (among patches) interact and affect eco-evolutionary dynamics in metapopulations (*Van Petegem et al., 2015*; *Jacob et al., 2015a*, *2017*, *2018*) and during

range expansions (*Fronhofer & Altermatt, 2015*), contributing to a broader understanding of spatial patterns in ecology.

## Movement differences between dispersers and residents, and their relationship with morphology

We have found significant variation in movement within genotypes, which was modulated by the genotype (significant genotype by dispersal status interaction): disperser cells within the same genotype moved faster and straighter than residents, suggesting different movement strategies, which were realised to different degrees by different genotypes. These differences are partly explained by cell morphology co-varying with movement. This is expected, as the energetic costs of movement of microscopic organisms in aquatic environments are heavily influenced by their morphology such as cell elongation and size (*Mitchell, 2002*; *Young, 2007*). Indeed, we found that larger cells moved faster, regardless of their dispersal status. The shape of the cells also influenced speed and linearity: dispersing cells that were more elongated moved faster and more linear, whereas resident cells did not show such a relationship. The differences in movement speed are likely due to different costs associated with motion in the liquid medium, with larger cells potentially having larger energy reserves and/or stronger movement machinery (*Mitchell, 2002*). This is corroborated by the fact that size always favoured faster movement, even when accounting for the genotype effect (see Fig. S4). Our results therefore closely agree with recent findings about a general allometric relationship between body size and movement speed (*Hirt et al., 2017a*, *2017b*).

We have shown that movement variation can be partly explained by different cell sizes and shapes. This is in line with previous findings on the condition dependence of dispersal that indicated that cell size and shape have an influence on the dispersal propensity (*Pennekamp et al., 2014*). However, in contrast to dispersal, larger and more elongated cells move faster and straighter, whereas more elongated and smaller cells disperse more. This contrasting result suggests that although larger cells may be superior in terms of movement ability, they may not disperse as much as expected as other causes of dispersal may be more important; for instance, dispersal decisions may be taken as a function of competitive ability rather than movement ability per se (*Fronhofer et al., 2015*). If cell size positively co-varies with competitive ability, smaller cells may disperse to escape the local competition although they have relatively weaker movement capabilities.

Aggregation behaviour of *T. thermophila* ciliates is another candidate for explaining movement differences because aggregation affects the spatial cohesion of a population and is a proxy for cooperative behaviour (*Schtickzelle et al., 2009*; *Chaine et al., 2010*; *Jacob et al., 2015b*). In a previous study, genotypes characterised by different degrees of aggregation did not show any relationship with dispersal (*Schtickzelle et al., 2009*). Instead aggregation co-varied with the occurrence of specialised dispersal morphs, which only appear during prolonged periods of starvation. Given the strong correlation we found between dispersal and movement, aggregation seems less likely to be a causal driver of the observed differences in movement, albeit information about cooperative strategies was found to influence dispersal decisions (*Jacob et al., 2015b*).

### Explaining dispersal rate with activity and movement variation

The amount of variation explained increased from 27% accounting only for genotype-specific cell activity level, to 37% when considering only genotype-specific movement speed, and up to 47% when considering genotype-specific activity and movement. Activity and movement hence provide complementary information about dispersal. For instance, in certain genotypes, individual cells may move faster and straighter, but their activity level may be lower, compared to a less mobile genotype were cells are generally more active. The increasing amount of variation explained in our study supports the claim of previous studies that behavioural differences are important for the correct prediction of large scale population distributions from small scale movement observations (*Morales & Ellner, 2002*; *Newlands, Lutcavage & Pitcher, 2004*). However, our results also indicate that other processes, including subtle behavioural differences among genotypes to enter narrow tubes, may contribute to the observed variation in dispersal. As the causes of movement and dispersal are not entirely known for each genotype in our study, both positive and negative influence on the genetic variation are plausible as one cause (e.g. density of conspecifics) may be more important for some genotypes than for others (*Pennekamp et al., 2014*).

### What are the consequences of the geno- and phenotypic variation in movement behaviour observed in our study?

Natural populations of *T. thermophila* ciliates are often constituted of multiple genotypes (*Doerder et al., 1995*), which may differ in movement behaviour as shown here. Modelling work has shown that communities/populations consisting of multiple phenotypes can actually show faster invasion speeds than that of the fastest monomorphic population alone (*Elliott & Cornell, 2012*). This was, however, only the case if the two phenotypes, that is a resident and a dispersive type, showed co-variation between growth rate and dispersal ability (e.g. well growing but poorly dispersing resident vs. poor growing and well dispersing establisher) and if the ratio between genotypes in these parameters varied 2–10 fold. Looking at the variation of our genotypes (Fig. 4), we see that the ratio in dispersal rate can be up to 10 fold depending on the genotypes contrasted. This suggests that with a known variation in growth rate with a factor of about two (*Pennekamp, 2014*), accelerating invasions of *Tetrahymena* are possible, if natural populations are more phenotypically diverse. Validating these predictions in experiments with mixed populations and their link with local adaptation would be a fruitful avenue for future research.

## CONCLUSIONS

Our study showed a close link between movement and dispersal on multiple levels. Dispersal predictions steadily improved when genotype differences in both activity level and movement behaviour were considered. This highlights that predictions of dispersal will benefit from a detailed understanding of the underlying movement behaviour. To move beyond short-term ecological predictions of dispersal dynamics, for example range expansions and range shifts due to environmental change, we would need to further

improve our understanding of how movement is affected by environmental variation and the relative fitness prospects of cells if staying in their current habitat patch or emigrating to another patch, which can lead to habitat choice, which has been shown in the species linked to temperature (*Jacob et al., 2017*, *2018*).

## ACKNOWLEDGEMENTS

Virginie Thuillier and Linda Dhondt provided valuable help during the experiment and data collection. F.P. Doerder kindly provided a collection of 22 wild type genetic lines of *T. thermophila*. We thank Delphine Legrand, Emanuel Fronhofer, Staffan Jacob, Camille Turlure, Justin Calabrese and anonymous reviewers for providing valuable comments on earlier drafts of the manuscript. This is publication BRC360 of the Biodiversity Research Centre.

### Funding

Frank Pennekamp was funded by Fonds Spéciaux de Recherche, Université catholique de Louvain. Nicolas Schtickzelle is a Senior Research Associate of the Fund for Scientific Research (F.R.S.-FNRS). Financial support was provided by F.R.S.-FNRS (PDR T.0211.19) and Université catholique de Louvain (ARC 10-15/031). Funding for Jean Clobert is provided by the Laboratoire d'Excellence (LABEX) entitled TULIP (ANR-10-LABX-41). The funders had no role in study design, data collection and analysis, decision to publish, or preparation of the manuscript.

### Grant Disclosures

The following grant information was disclosed by the authors:
Fonds Spéciaux de Recherche, Université catholique de Louvain.
F.R.S.-FNRS: PDR T.0211.19.
Université Catholique de Louvain: ARC 10-15/031.
Laboratoire d'Excellence (LABEX) entitled TULIP: ANR-10-LABX-41.

### Competing Interests

Jean Clobert is an Academic Editor for PeerJ. The authors declare that they have no competing interests.

### Author Contributions

- Frank Pennekamp conceived and designed the experiments, performed the experiments, analysed the data, contributed reagents/materials/analysis tools, prepared figures and/or tables, authored or reviewed drafts of the paper, approved the final draft.
- Jean Clobert conceived and designed the experiments, contributed reagents/materials/ analysis tools, authored or reviewed drafts of the paper, approved the final draft.
- Nicolas Schtickzelle conceived and designed the experiments, analysed the data, contributed reagents/materials/analysis tools, prepared figures and/or tables, authored or reviewed drafts of the paper, approved the final draft.

## Data Availability

The following information was supplied regarding data availability: The data and code to reproduce the analyses are available at figshare: Pennekamp, Frank (2019): Data from: The interplay between movement, morphology and dispersal in Tetrahymena ciliates. figshare. DOI 10.6084/m9.figshare.5882530.v1.

Pennekamp, Frank (2019): Code from: The interplay between movement, morphology and dispersal in Tetrahymena ciliates. figshare. DOI 10.6084/m9.figshare.5882635.v1.

## Supplemental Information

Supplemental information for this article can be found online at http://dx.doi.org/10.7717/peerj.8197#supplemental-information.

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
