# Peer review of "The interplay between movement, morphology and dispersal in Tetrahymena ciliates"

_PeerJ, doi:10.7717/peerj.8197_

## Round 0.1 · original submission · Major Revisions

Both reviewers had similar comments on your analytical methods, indicating that more/different analyses need to be carried out.

Reviewer 1 ·

Basic reporting

Mostly the text is very clear. The Figure and Table legends require more information. For instance, describing what the different symbols refer to.

Experimental design

The experimental design is sound and clearly described. I found the description of the statistical analysis and results for the second question very difficult to follow. Please see my comments to the author.

Validity of the findings

no comment

Additional comments

This study investigates the genetic and environmental link between movement, morphology and dispersal across 44 Tetrahymena genotypes. First they describe how movement behaviour, morphology and dispersal is linked within and across the 44 different genotypes, before quantifying how much variation in dispersal can be attributed to these differences in morphology and movement. They show that relationships between these traits have both a genetic and environmental component. Some genotypes had a greater tendency to disperse, however, across all genotypes there was a difference in morphology between dispersers and residents.

Overall I enjoyed reading this manuscript and recommend it for publication in PeerJ. My major concern is that I found the description of the analyses and results for question 2 (link between movement and morphology) very difficult to follow. Some things became clear from simultaneously looking at the figures and supplemental Tables, but this information should be clear from the main text. Please see my more detailed comments below.

Lines 54 – 56: maybe add that dispersal movements are necessarily in sub-divided populations.

Line 73: Add reference for ‘environmental causes’.

Line 84: is there a general effect of body condition on dispersal across species, or is it species and context specific?

Lines 91 – 104: this paragraph logically follows on from line 80. Also, on first reading I thought this contradicted the information presented in lines 76 – 80. Before you state there is new technology to investigate dispersal, then here you state difficult to track long-distance dispersal. But on a second reading I see that lines 91 – 94 are talking about genetic effects and before (lines 76 – 80) the genetic component could not be disentangled. This could be stated more explicitly.

Line 104: you could state here what the next step needs to be. A study across lots of genotypes which is easily done with microcosm experiments.

Lines 112 – 121: Changing the order of these sentences might provide a better context for your study. I’d put the last sentence first and change it to say ‘There is compelling evidence that dispersal in this organism………’, then the second sentence describing all these studies and finally say ‘There is extensive variation among genotypes……’. Then you could say how the present study advances on Fjerdingstad et al. 2007 and Pennekamp et al. 2014.

Lines 122 – 131: Describe Fjerdingstad et al. 2007 and Pennekamp et al. 2014 in more detail. Were the same genotypes used? And then frame the questions so that it is clear how this study is different.

Line 152: add the length of the tube.

Line 161: add that there were 5 photos taken, 1 for each chamber. You could add the text from lines 60 – 64 of the supplemental material.

Line 177: ….all the….

Lines 192 – 197: It is not clear how this measure also integrates step length and turning angle?

Line 201: The term ‘dispersal system’ is introduced out of the blue. State earlier on that your 2 patch population (or maze) is referred to as a ‘dispersal system’.

Lines 231 – 237: This is quite confusing. Change ‘condition-dependence of movement’ to ‘to see if there were differences between residents and dispersers’. It is unclear how the models were constructed. Was movement speed and tortuosity measured separately in 2 models each with size and shape as fixed factors? From the supplemental material it looks like there was one model for size and another for shape on each explanatory variable. This should be stated. Why not do one model with both size and shape? How did you account for differences in dispersal status in these models (lines 233 – 234)? From the supplemental material it looks like ‘start’ or ‘target’ tube was included in models. This should also be stated. And then were models repeated with the mean value for each genotype (lines 235 – 236)? If this is the case were means for genotype calculated as the mean for each replicate and then the mean of these two replicates?

Line 253: This contradicts the previous sentence. Maybe say in the previous sentence that for this was the result for majority of genotypes.

Lines 264 – 266: this sentence doesn’t make sense. Table S4 shows this to be the most parsimonious model ‘log(FractalD - 1) ~ size + tube + size:tube + 1’, which to me means that there is a significant interaction between size and disperser/resident for tortuosity.

Lines 270 – 276: this is really confusing.

Lines 293 – 295: The end of this sentence is unclear.

Lines 317 – 321: I don’t completely follow the logic here. Is there any evidence that your movement behaviours are linked to foraging? And why only in non-territorial animals?

Line 322: change ‘currently receives more attention’ to ‘has recently received more attention’ and you could describe this study in more detail.

Lines 360 – 364: this sentence is confusing.

The tables in the supplementary material (Table S2 – S13) are confusing. What is tube in these models? Each Table should have a legend at least stating what the response variable was? Is the model in bold the one which was retained?

Legends should be included for the figures in the supplementary materials?

Figure 2: are the circles and triangles dispersers and residents? This information should be in the figure legend.

Figure 3 looks like it shows the means for replicates and dispersers for each tube. Was this the response variable in the analyses described in lines 231 – 237?

Table 1: add the denominator degrees of freedom.

·

Basic reporting

This is all fine.

Experimental design

The experimental design is sound and the questions are interesting and well defined.

Validity of the findings

The analyses have serious shortcomings that call the validity of the results into question. Please see my "general comments for the author" for full details on the aforementioned problems and potential solutions to them.

Additional comments

Review of “The interplay between movement, dispersal and morphology in Tetrahymena ciliates” by Pennekamp et al.

This paper uses a tractable lab system coupled with elegant experiments to tease apart relationships between movement, dispersal, genetics, and morphology. I think the questions the authors ask are timely and interesting, and this type of experimental system can help fill gaps between theory and the type of data it is possible to collect under field conditions. So from that perspective, I think this study could be a useful addition to the literature. However, I have some serious issues with the analysis of the movement data that I document at length below. I am signing my review in the interest of full disclosure, as the some of the solutions I propose are based on my own work on movement analysis. I have tried to give careful, objective justifications for the both the problems I raise and the solutions I propose, but wanted to give the authors full information with which to arrive at their own conclusions.

The analyses in this paper are all based on Turchin's 1998 book. To be clear, I think Turchin's book was a landmark in the history of movement ecology and was state-of-the-art for its time. However, huge advances in movement analysis have been made in the 20 years since Turchin's book was published, and none of that progress is leveraged this manuscript. This is especially troubling given that the type of data the authors have collected, with extremely high frequency sampling, represents a worst case scenario for the discrete-time RW methods described in Turchin.

I think the authors would benefit greatly from a shift in perspective about the nature of the data they have collected. The statements about “spurious autocorrelation” (e.g., Ln 181) in the data and the need to remove it, while understandable, are misguided. There is nothing spurious about autocorrelation in movement data. Autocorrelation is a fundamental property of movement data, and the more finely you sample, the more autocorrelation will be revealed. Indeed, any movement process that obeys the laws of physics must feature autocorrelated positions, velocities, accelerations, etc... (E.g., truly uncorrelated positions imply that the animal would be capable of infinite velocity, truly uncorrelated velocities imply that the animal would be capable of infinite acceleration, etc...). The only question is “did you sample finely enough to reveal the inherent autocorrelation in the data”? Furthermore, the precise nature of the autocorrelation in the data contains a wealth of information about the underlying movement process that is being studied. In other words, the movement process leaves a telltale signature in the autocorrelation structure of the data, but decoding that information requires suitable methods.

The discrete-time RW model has a number of severe limitations that render it an inappropriate basis for analysis of high-frequency, highly autocorrelated movement data. This model can only accommodate a very limited form of autocorrelation, and also depends sensitively on the sampling coinciding with behaviorally important events. A consequence of the need to match sampling to behavioral events is that parameter estimates and inferences under this model are highly scale dependent, meaning that if you sample the exact same path at two different sampling resolutions, you will get (potentially very) different parameter estimates. The root of the problem here is that discrete-time RW models confound the (usually arbitrary) sampling process with the underlying (and usually unknown) movement process. In practice, dealing with these limitations means that the investigator must know exactly when to take a sample (i.e., every time the tracked individual makes a movement decision), and if you get it wrong, the inherent scale-dependence of the model guarantees that you will get biased results. In the vast majority of realistic cases including that in the present manuscript, one simply does not know and cannot know the perfect sampling strategy that would result in accurate inferences from this type of RW model. Any time I see statistical methods that only work when the sampling is perfectly right (i.e., require “Goldilocks” sampling, not too fine, not too coarse), I take that as a clear sign that there is something fundamentally wrong.

So the problem here is not your data (which should be good news!), but instead that the model you're trying to use to analyze it is badly misspecified for the data you have. Back when Turchin's book was written, this approach, and the ad-hoc data manipulation it required to shoehorn real movement data into it (like the Douglas–Peucker thinning the authors did), was really the only option. It is also worth noting that the thinning method employed here, if I understood it correctly, would thin each path differently, which when coupled to the inherent scale dependence of discrete-time RWs, would result in *differential* biases among individuals, making any comparisons across individuals or groups hard to interpret. Like I said, however, the game has changed a lot since Turchin's book, and there are now much better methods. The problems with the old approach have also become more apparent given that technologies like the video capture techniques used here, and modern GPS devices, allow for much finer sampling than was possible 20 years ago. The solution is to formally separate the discrete-time sampling process, from the underlying continuous-time movement process that is being sampled. Real animals *always* trace continuous paths through the environment, meaning that the animal had to be somewhere at any given point in time. Turchin acknowledges this fundamental truth at the beginning of the section (5.2.1, pg. 128) on undersampled or oversampled paths.

Continuous-time stochastic process (CTSP) movement models formally accomplish this separation of sampling from movement process, and this is where the link to my own work comes in. While this separation sounds subtle, it changes things fundamentally. CTSPs do not require any ad-hoc manipulation of the data, they naturally handle gaps and missing data, and they do not require matching of the sampling to unknown behavioral processes. Their parameter estimates are scale-independent, with the caveat that to resolve a process operating on timescale tau, the sampling interval must be < tau, which only means that the models are not magic and cannot “see” below the resolution of the data. A key advantage of these models is that they can naturally accommodate arbitrarily autocorrelated data, again with no need to thin or otherwise manipulate the data. A catalog of such models currently exist, and they can be tailored to a give dataset via model selection.

From the description of your data and the sampling frequency you used, I would guess the model that would be most appropriate for your data would be the Integrated Ornstein-Uhlenbeck (IOU) process. This model is also sometimes referred to as the “continuous velocity model” (by Gurarie and colleagues) or the “continuous-time correlated random walk model” (by Johnson and colleagues). The IOU process features autocorrelated positions and autocorrelated velocities (i.e., directional persistence in the movement), and thus should be able to handle your complete datasets. A parameter of this model called the “velocity autocorrelation timescale” can be used to quantify path tortuousity. It measures the typical amount of time it takes for directional persistence (=velocity autocorrelation) in the movement to decay, with a small value indicating a more tortuous path, and a larger value indicating more ballistic-like movement. It is necessarily estimated when this model is fit to the data, and the estimates come equipped with confidence intervals to quantify their uncertainty. A fitted IOU model can also be used as the basis for a scale-independent, and more accurate estimate of movement speed, which is also accompanied by confidence intervals. (Note that summing straight-line distances between location observations and dividing by the total time elapsed, as was done in this study, gives a lower bound on speed, as it assumes the shortest possible travel distance. Additionally, finite difference estimators like this are also highly scale dependent, do not come with confidence intervals, etc...). With those two pieces, you should be able to perform a much more appropriate and robust version of your analysis. I would recommend dropping the diffusion coefficient analysis, as it doesn't contain any information that is not already captured by the speed and tortuousity analyses.

As for how to do this, there are basically two options: the continuous-time movement modeling (ctmm) R package that my lab develops, or the continuous-time correlated random walk library (crawl) R package by Devin Johnson (not affiliated with me or my group in any way). I would recommend using ctmm, but my bias there should be obvious. I do think ctmm has 3 advantages for what the authors want to do relative to crawl: 1) ctmm includes a library of candidate models including IOU, whereas crawl implements only the IOU model. As I said, I would guess that IOU is what you want for your data, but with ctmm, you could formally check if it is indeed best via model selection. 2) both packages will produce velocity autocorrelation timescale estimates and speed estimates, both with CIs, but ctmm's speed() function uses a more sophisticated and accurate estimator than crawl. 3) We frequently use ctmm as a teaching tool in the AniMove movement analysis courses, and we get lots of feedback from students on usability, which we've repeatedly used to refine the interface and workflow. Crawl is definitely functional but is less refined. Either approach, however, would be a substantial improvement over the current analysis, and would allow the authors to make valid comparisons among the different groups in the study. For ctmm, Calabrese et al. (2016, in Methods in Ecology) gives an overview of the package and workflow. The speed() function is available with documentation in the most recent version of the package, but the paper describing the methods behind it has not yet been published (I'm happy to share the ms). Ctmm also has a number of detailed vignettes that walk a new user through how to use it. A full list of papers on the technical methods that ctmm implements can be found at http://biology.umd.edu/movement.html. For the crawl package, Johnson et al. (2008 in Ecology) describes the model and underlying estimation methods. Both packages can be found on CRAN.

Sincerely,

Justin Calabrese

---

## Round 0.2 · Minor Revisions

Please follow the remaining minor comments and suggestions of the reviewer in your revised version.

·

Basic reporting

This is fine.

Experimental design

This is fine.

Validity of the findings

See comments below.

Additional comments

I thank the authors for taking my comments seriously and re-doing the analysis based on continuous-time models. I recognize that this was a lot of work, so again thank you. I believe that these efforts will pay off in the long run, as this paper will set a precedent for the analyses of these kinds of datasets.

I am fine with the author's use of the Smoove package and can see that it is setup in a more convenient way for what you wanted to do. I will note, however, that the apparent speed advantage the authors mention as part of their justification for using Smoove over ctmm (or crawl) comes at a cost. ctmm and crawl implement only exact model fitting methods, which are statistically more rigorous and more reliable, but slower. The default fitting method in Smoove, "velocity likelihood (vLike)", is an approximation that first calculates velocity estimates by differencing the position data, and then treats those estimates as though they were data (i.e., as though velocity had been directly observed by the tracking device). That approach works well when the velocity estimates are good, but can become inaccurate if they are not good. There are many real-world cases where such finite difference velocity estimates can be badly biased, so this is definitely a "user beware" kind of situation, and I believe Smoove should make that clearer to the user. Although I suspect this approximation is ok in your case, given how finely sampled your data are in time, I would like to see an acknowledgement in the paper that an approximate fitting method was used. I would also like to see a spot check on the approximate method added to the appendix. To check if the approximation is ok, you should select a random subset of datasets on which you fitted models via vLike, and do the fit again using the option method='crawl'. This will use the exact likelihood function for the full position data (what Gurarie calls "zLike"), but uses the fast implementation of that likelihood it in the crawl package. Smoove does not implement fast algorithms for the exact likelihood itself, and so has to rely on crawl's fast implementation of this method. (Incidentally, the fast algorithm used for fitting the exact position likelihood in crawl is equivalent to the method ctmm uses for the same model.) You should then compare the two different sets of estimates (exact vs approximate) for each dataset you spot checked. If they differ substantially, you should be concerned, as the exact results are more reliable. I think a small and truly randomly selected subset of your 6072 analyzed trajectories (say, 50 in total) would suffice. This should be added to the supplementary material as a check on the quality of the approximate fits.

Minor comments

There is still mention of "diffusion rates" in the abstract, even though the results on diffusion rates have been removed from the manuscript.

r.m.s. does not stand for "random movement speed" as mentioned on ln 213 in the text. This parameter is the "root mean square" (hence r.m.s) speed.

Also, the "characteristic timescale of autocorrelation" on the next line should be referred to as the velocity autocorrelation timescale, as it specifically relates to the timescale over which autocorrelation in velocity decays. This is an important clarification, as some closely related movement models have multiple autocorrelation scales (position autocorrelation, velocity autocorrelation, etc.).

Sincerely,

Justin Calabrese

---

## Round 0.3 · accepted · Accept

Dispersal, in the broader sense of the term, is the movement away from the birthplace and is a common phenomenon that occurs in most living organisms at some point of their life history. The genotypic basis in movement behavior of ciliates is an important finding.